# Evaluation of the Sensititre YeastOne and Etest in Comparison with CLSI M38-A2 for Antifungal Susceptibility Testing of Three Azoles, Amphotericin B, Caspofungin, and Anidulafungin, against *Aspergillus*
*fumigatus* and Other Species, Using New Clinical Breakpoints and Epidemiological Cutoff Values

**DOI:** 10.3390/pharmaceutics14102161

**Published:** 2022-10-11

**Authors:** Marcia S. C. Melhem, Vivian C. Coelho, Claudia A. Fonseca, Lidiane de Oliveira, Lucas X. Bonfietti, Maria. W. Szeszs, Marcello M. C. Magri, Francine S. Dorneles, Hideaki Taguchi, Daniel V. S. Moreira, Adriana L. Motta, Marjorie V. Batista, Katsuhiko Kamei, Maria A. Shikanai-Yasuda

**Affiliations:** 1Mycology Unit, Parasitology and Mycology Department, Instituto Adolfo Lutz, Secretary of Health, Government of São Paulo State, São Paulo 01246-902, SP, Brazil; 2Graduate Program in Sciences, Coordination of Diseases Control, Secretary of Health, Government of São Paulo State, São Paulo 01246-902, SP, Brazil; 3Graduate Program in Infectious and Parasitic Diseases, Faculdade de Medicina, Federal University of Mato Grosso do Sul, Campo Grande 79070-900, MS, Brazil; 4Laboratório de Investigação Médica em Imunologia (LIM 48), Hospital das Clínicas, Faculdade de Medicina, University of São Paulo, São Paulo 05403-000, SP, Brazil; 5Division of Infectious Diseases, Hospital das Clínicas da Faculdade de Medicina, University of São Paulo, São Paulo 05403-000, SP, Brazil; 6Medical Mycology Research Center, Chiba University, Chiba 260-8673, Japan; 7Laboratory of Microbiology, Division of Central Laboratory—Laboratory of Medical Investigation—LIM 03, Hospital das Clínicas, Faculdade de Medicina, University of São Paulo, São Paulo 05403-000, SP, Brazil; 8Department of Infectious and Parasitic Diseases, Faculdade de Medicina, University of São Paulo, São Paulo 05403-000, SP, Brazil

**Keywords:** gradient diffusion assays, *Aspergillus fumigatus*, Etest, CLSI, Sensititre YeastOne, azole antifungals

## Abstract

Aspergillosis is an invasive fungal disease associated with high mortality. Antifungal susceptibility testing (AFST) is receiving increasing consideration for managing patients, as well as for surveilling emerging drug resistance, despite having time-consuming and technically complex reference methodologies. The Sensititre YeastOne (SYO) and Etest methods are widely utilized for yeasts but have not been extensively evaluated for *Aspergillus* isolates. We obtained Posaconazole (POS), Voriconazole (VCZ), Itraconazole (ITC), Amphotericin B (AMB), Caspofungin (CAS), and Anidulafungin (AND) minimum inhibitory concentrations (MICs) for both the Etest (*n* = 330) and SYO (*n* = 339) methods for 106 sequenced clinical strains. For 84 *A. fumigatus,* we analyzed the performance of both commercial methods in comparison with the CLSI-AFST, using available cutoff values. An excellent correlation could be demonstrated for Etest-AMB and Etest-VCZ (*p* < 0.01). SYO-MICs of AMB, VCZ, and POS resulted in excellent essential agreement (>93%), and >80% for AMB, VCZ, and ITC Etest-MICs. High categoric agreement was found for AMB, ITC, and CAS Etest-MICs (>85%) and AMB SYO-MICs (>90%). The considerable number of major/very major errors found using Etest and SYO, possibly related to the proposed cutoffs and associated with the less time-consuming processes, support the need for the improvement of commercial methods for *Aspergillus* strains.

## 1. Introduction

The incidence of invasive aspergillosis has increased considerably in the past few decades. Many factors have contributed to this increase such as the increasing number of patients who undergo organ transplants and corticosteroids therapy, and the spread of the COVID-19 pandemic [1,2,3]. The disease is associated with high mortality and antifungal susceptibility testing (AFST) has been receiving increasing consideration as a valuable tool for managing patients as well as for epidemiological surveillance of emerging drug resistance [4]. The Clinical and Laboratory Standards Institute (CLSI) and European Committee on Antimicrobial Susceptibility Testing (EUCAST) have established standard procedures for testing the susceptibility of most prevalent molds and yeasts of clinical relevance to antifungal agents and have proposed some species-specific clinical breakpoints (CBPs) for interpreting MIC results of some antifungal agents [4]. AFST plays a significant role in patient management by aiding the prescription of an appropriate antifungals; however, the reference methodologies are time-consuming and technically complex, leading laboratories to adopt commercially available alternatives [5].

An increasing number of hospital microbiology laboratories are performing AFST. The medical demand for timely AFST results prevents the routine use of the laborious reference broth microdilution method and promotes interest in the available commercial AST systems. An alternative is the use of gradient diffusion strips, Etest being one of the most adopted worldwide. In this context, the method’s performance should be extensively evaluated using clinical isolates from all over the world. Etest has acted as a valuable alternative for the detection of emerging non-wild-type (non-WT) resistance as epidemiological cutoff values (ECVs) for this method have been recently defined for several drug-species combinations. Otherwise, studies evaluating the agreement between the Etest method and the reference methods for filamentous fungi showed variable results depending on the antifungal agent, species, and incubation time [4,6,7].

In addition, the colorimetric broth microdilution SYO panel is widely utilized for the AST of *Candida* in the clinical laboratory and less utilized with *Aspergillus* isolates for research purposes [8,9]. Previous data were limited by the small number of tested *Aspergillus* clinical isolates from the South American region, thus precluding an evaluation of Etest’s role for determining local resistance.

While susceptibility testing with antifungal agents against molds such as *A. fumigatus* isolates has been determined extensively in reference methods, it has not been widely evaluated by commercial methods. The role of AFST systems in the detection of resistance in *A. fumigatus* has not been extensively assessed as few isolates have been tested to date [8,10,11,12]. Therefore, we chose to assess the Etest and SYO’s ability in performing AFST for 106 clinical isolates of molecularly identified *Aspergillus* clinical isolates.

For commercial methods, there are no suitable clinical data to distinguish susceptible or resistant isolates, and available epidemiological cutoff points were applied to interpret the MIC results. We then used the Etest method and SYO panel to determine antifungal agents’ MICs for the isolates to calculate correlations, and essential and categorical agreements with the reference broth microdilution method according to the CLSI M38-A2 document [13].

Species-specific CBPs for interpreting MIC results serve as predictors of clinical success of treatment. CLSI has established CBPs for the more prevalent *Aspergillus* species, *A. fumigatus,* and VCZ in the M59-ED3 document (CLSI M59-ED3, 2020) [14]. In the absence of BPs, ECVs should identify non-WT isolates (MIC ≤ ECV) with reduced susceptibility to the drug under analysis [15]. We identified errors in both commercial methods for determining the profile of susceptibility. This enabled us to advise laboratories to use Etest and SYO systems for azole-MIC and Amphotericin B-MIC and to prevent them from using SYO on a routine basis for testing echinocandins against *Aspergillus* isolates.

## 2. Materials

The highest possible commercially available grade of Itraconazole (ITC), Voriconazole (VCZ), Amphotericin B (AMB), Caspofungin (CAS), and Anidulafungin (AND) was purchased from Sigma-Aldrich (St. Louis, MO, USA). The organic solvent for the drugs, dimethyl sulfoxide in analytical grade, was obtained from Sigma (St. Louis, MO, USA). Further drug dilutions were made in Roswell Park Memorial Institute (RPMI) 1640 with L-glutamine, without bicarbonate, buffered with 0.165 morpholine propane sulfonic acid to pH 7.0 (RPMI; Sigma Chemical Co., St. Louis, MO, USA). Etest strips containing ITC, VCZ, AMB, CAS, and AND were used (bioMérieux, Marcy L’Etoile, France). Sensititre YeastOne YO10 was purchased from TREK (Diagnostic Systems Ltd., West Sussex, UK). The fungal conidia inoculums were prepared in RPMI 1640 Medium (Sigma, USA) supplemented with glucose (2% final concentration). Potato dextrose agar (PDA; Becton Dickinson, Sparks, MD, USA), Bacto agar (Difco Laboratories, Detroit, MI, USA), and Sabouraud dextrose agar (SDA; Becton Dickinson, Sparks, MD, USA) were used in the study.

## 3. Methods

### 3.1. Aspergillus Species Identification

Isolates were received from the Laboratory of Microbiology—Division of Central Laboratory (Laboratory of Medical Investigation—LIM 03) of the Hospital das Clínicas, Faculdade de Medicina, University of São Paulo (HC-FMUSP), Brazil, and were originally recovered from 106 patients of HC-FMUSP, Brazil. This laboratory is accredited by the American College of Pathology and follows all that provider’s guidelines for performing fungal cultures.

The isolates were identified as 84 *A. fumigatus,* 9 *A. niger,* 7 *A. flavus,* 3 *A. clavatus,* 1 *A. terreus,* 1 *A. awamori,* and 1 *A. welwitschiae* by the Laboratory of Medical Investigation in Immunology (LIM 48) of HC-FMUSP, Brazil.

For molecular identification, the segment of the b-tubulin gene was amplified using primers bT2a and bT2b [16]. Similarly, a region of the rodlet gene was amplified using the primers rodA [17]. The calmodulin gene was also amplified as previously described [18]. The ITS regions (ITS-1 and ITS-2) of the ribosomal RNA (rRNA) gene complex were amplified [19] when the other gene fragments were discordant with each other. The sequences obtained were compared with sequences deposited in the GenBank (http://www.ncbi.nlm.nih.gov/BLAST (accessed on 15 April 2015). DNA sequences representing ITS regions, b-tubulin, and calmodulin genes were aligned using ClustalX and visually edited in the Genedoc version 2.6. MEGA (MAC version 6) program to generate and edit the phylogenetic trees, and the similarity/dissimilarity amongst the sequences of various Aspergilli was studied.

### 3.2. In Vitro Antifungal Susceptibility Testing

AFST by CLSI and Etest and SYO methods were initially performed at the Laboratory of Medical Investigation in Immunology (LIM 48), HCFMUSP, and 15% were also analyzed by CLSI at the Medical Mycology Research Center, Chiba, Japan. All isolates were submitted from the LIM to the Reference Laboratory Adolfo Lutz Institute (IAL, São Paulo, Brazil) for antifungal susceptibility duplicated testing through CLSI (*n* = 106) and Etest (*n* = 84) methods. SYO was performed at the Laboratory of Medical Investigation (LIM 48), HCFMUSP, and partially (17%) in duplicate at the Infectious and Parasitic Diseases Laboratory (LabDIP) of the Federal University of Mato Grosso do Sul (MS, BR).

Two quality control isolates (QC), *Candida krusei* ATCC 6258 and *Candida parapsilosis* ATCC 22019 were included as controls in all experiments (CLSI M61-ED2, 2020) [20].

#### 3.2.1. CLSI M38-A2 Methodology

The AFST was performed as outlined in CLSI document M38-3rd Ed. (CLSI. Reference Method for Broth Dilution Antifungal Susceptibility Testing of Filamentous Fungi. 3rd ed. CLSI standard M38. Wayne, PA: Clinical and Laboratory Standards Institute; 2017) [13]. Briefly, the susceptibilities of the isolates to ITC, VCZ, POS, AMB, CAS, and AND were assayed by the broth microdilution method. Isolates were grown on potato dextrose agar (PDA) at 35 °C up to 48–72 h to maximize conidial harvest, and the conidia were counted with a Neubauer chamber and adjusted to a concentration of 10^6^ CFU/mL. The microdilution was performed with twofold dilutions of the drugs at concentrations ranging from 0.03 to 16 mg/mL. The triazole-MIC and AMB-MIC were defined as the lowest drug concentrations that showed a complete reduction in fungal growth. For CAS and AND, the minimal effective concentration (MEC) was defined as the minimum concentration of drug that produced abnormal hyphal growth with highly branched tips. Susceptibilities were determined by duplicate measures in the LIM 48 and IAL laboratories.

We first determined the MIC of the routinely used drugs in cases of Aspergillosis, namely three triazole agents and the polyene Amphotericin B as well as two echinocandins (Caspofungin and Anidulafungin) against 106 clinical strains of *Aspergillus* spp. The MIC results were then assessed to classify the isolates into different susceptibility categories, according to the existing clinical breakpoints for each drug-species pair.

#### 3.2.2. Etest Method

Etest (biomérieux, Marcy-l’Étoile, France) consists of a predefined gradient of antifungal agent concentrations on a plastic strip which, after placement and incubation in an inoculated agar plate, results in an ellipse of growth inhibition that is used to determine the MIC of the drug being tested. Numerous previous studies have examined the susceptibility of yeasts by the Etest gradient strip method with consistent excellent results, as reviewed by Espinel-Ingroff, 2022 [21], but data on its performance with molds are still lacking [22].

All the isolates were subcultured for three to four days on PDA at 35 °C before testing. Inoculum of up to 106 isolates of conidial suspensions were successively prepared in sterile saline, adjusted to a concentration of 10^6^ CFU/mL (68 to 82% transmittance at 530 nm), except for *A. nigri* isolates suspensions, which were counted with a Neubauer chamber, adjusted to a concentration of 10^6^ CFU/mL, and swabbed onto RPMI 1640 agar in three directions. The agar version of the medium was obtained by using RPMI 1640 medium solidified with 1.5% Bacto agar. The strips of ITC, VCZ, AMB, CAS, and AND (concentrations ranging from 0.002 to 32 mg/L) were applied to the inoculated agar. The MIC was determined at 100% inhibition for all tested antifungals at the interception of the elliptical growth inhibition halo to the scale of the antifungal strip. MICs were determined after 24 and 48 h of incubation at 35 °C.

#### 3.2.3. SYO Panel

The MICs of POS, VCZ, ITC, AMB, CAS, and AND (concentrations between 0.0015 and 8 mg/L) were determined at 100% inhibition for all tested antifungals and read at the lowest drug concentration that produced a color change. The growth medium contained resazurin, an indicator of cell viability that turns from blue to pink when oxidized by viable fungi. Inoculum concentration was adjusted at a McFarland standard of 0.5 (0.5–5 × 10^6^ UFC/mL), except for *A. nigri* suspensions that were counted with a Neubauer chamber.

MICs were determined after 24 h of incubation as the lowest antifungal concentrations at which the wells remained blue (no growth) and interpreted according to the CLSI breakpoints [23].

MICs were confirmed in different experiments executed in both the LIM 48 and LabDIP laboratories. Unfortunately, approximately 2/3 of the samples analyzed by SYO for echinocandins showed growth in all tested concentrations (up to 8 mg/L), while for the control isolates the expected MIC of CAS, AND, and micafungin were achieved. MIC values ≥ 8 mg/L were not registered for this study. The data presented for SYO refers to seven isolates of *A. fumigatus* and one isolate of *A. flavus*.

### 3.3. MIC/MEC Breakpoints

The antifungal susceptibility of all isolates was determined according to the current available CLSI CBPs or ECVs (CLSI M59-ED3, 2020) [14,24]. CBPs were divided into resistant and susceptible isolates of *Aspergillus* according to certain species and antifungal agents. ECVs were applied to classify isolates non-WT with decreased susceptibility and having probable resistance mechanisms, and wild-type (WT) isolates were defined as those that do not harbor any acquired resistance to the drug being examined [14,25].

Regarding *A. fumigatus* and AND, neither CBP nor ECV was defined by CLSI or elsewhere. For the remaining drugs, we adopted the existing CBP or ECV values as shown in Table 1 below.

### 3.4. MIC Data Analysis

All experiments were carried out in duplicates, or triplicates, and results were indicated as the modal value when distinct values were found. Etest MIC endpoints were raised to the nearest twofold dilution value that matched the CLSI concentration ranges to facilitate comparisons of results. MIC ranges were obtained for each species-drug combination by each method tested. The MIC_50s_ and MIC_90s_, which represent MICs at which 50% and 90% of the isolates tested are inhibited, respectively, were determined for species for which at least seven isolates were available.

Differences between MIC values of no more than two log_2_-dilutions were used to calculate the percentages of essential agreements (EAs) between Etest and CLSI and SYO and CLSI. Essential agreements of ≥90% between the two methods were considered acceptable [24]. These analyses were performed for results obtained with *A. fumigatus* due to the robust number of isolates.

Categorical agreements (CAs) between the susceptibility category of each isolate according to method dependent ECVs were calculated. Errors were categorized as very major errors (VMEs) or false susceptible when the commercial methods classified an isolate as susceptible/wild type for a given agent and the CLSI reference method classified it as resistant/non-WT. They were categorized as major errors (MEs) or false resistance when an isolate was classified as non-WT by commercial method and susceptible/WT by the CLSI gold standard method. A result was deemed to be a minor error (MiE) when, for a given agent, it was classified as wild-type/non-wild-type by any of the commercial methods studied but was determined to be intermediary by the reference CLSI method.

### 3.5. Statistical Analysis

The correlation among the susceptibility methods was determined by Pearson coefficients. Fisher’s exact test was used to determine the association between the CAs, according to the methods. The statistical analyses were performed using the Stata^®^ program (version 11.0, Stata Corp. LP, College Station, TX, USA). A *p*-value of <0.05 was considered significant. All graphs have been generated and analyzed using Prism nonlinear regression software (GraphPad Software, San Diego, CA, USA).

## 4. Results

### 4.1. MIC Results

The CLSI-MICs for the QC strains were within the recommended 24-h MIC/MEC limits (Appendix A).

According to the available CBP and ECV in tests with *A. fumigatus,* we included in this study one VCZ resistant isolate and four non-WT isolates for POS (*n* = 1), for CAS (*n*= 1), and for ITC (*n* = 2). For AMB, all isolates were of the wild-type.

Summarized in Table 2 are the MIC ranges (MIC_50_ and MIC_90_ values) of the six antifungal drugs tested against 106 isolates of *Aspergillus* spp. determined by the CLSI M38-A2, Etest, and for up to 58 isolates by colorimetric SYO methods.

### 4.2. Agreement of the Etest and Sensititre YeastOne with CLSI Reference Method for A. fumigatus Isolates

The essential agreement within ±2 Log_2_ dilutions for the comparison of Etest or SYO with the CLSI reference broth dilution method and categorical agreement results, when applied, are shown in Table 3.

All agreements were over 82.1%, except for SYO-ITC (78.3%), Etest-AND (66.7%), and Etest-CAS (35.3%).

For AMB, the EA, as well as CA, were high in both commercial methods, ranging from 82.1% to 95.2%.

For VCZ, the EA of SYO was superior (95.7%) to the EA of Etest (87.3%) compared with the CLSI results. Considering the CBP for VCZ, a single VCZ-resistant *A. fumigatus* isolate was correctly identified as non-WT by Etest and by SYO. In the identification of VCZ-intermediary isolates (*n* = 4), we verified more correct results (non-WT classification) by the SYO method (75%) than by Etest (33.3%). Among 41 susceptible isolates, Etest barely categorized 14.9% (4 out 27 tests) as wild-type category, similar to SYO (22%; 9 out 41).

For testing ITC by Etest, we verified a good EA (82.8%) and CA (90.6%). As for SYO, a single significant correlation was found between SYO and CLSI (r = 0.2958; *p* = 0.0460) for ITC MICs.

The EA found for SYO testing of POS was high (93.5%). Considering the proposed ECV, SYO was able to identify the single non-WT-POS isolate. However, it identified only 5 (11.1%) out of 45 WT-POS tested isolates, resulting in low CA (13.4%) due to the high amount of ME (93.5%).

## 5. Discussion

We determined the Etest MICs for antifungal agents used as first-choice and salvage therapy for invasive Aspergillosis. We additionally categorized the 84 isolates of *A. fumigatus* according to the proposed ECVs or CBPs of VCZ [14,20]. To our knowledge, most studies focus on *A. fumigatus* and evaluate MICs obtained by commercial methods with CLSI M38 methodology as a reference, and therefore, preclude robust comparison of our MIC findings with other *Aspergillus* species.

We will first discuss the MIC results of VCZ and AMB against *A. fumigatus* since both drugs are the backbone of antifungal therapy in cases of invasive Aspergillosis and information on in vitro susceptibility of *Aspergillus* clinical isolates to those drugs is relevant for validating CBPs and ECVs. In suspected or proven azole-resistant *A. fumigatus* cases, AMB remains the first-line therapy, and a reliable simple commercial AST is needed to provide the fungal susceptibility profile in a timely way to help with therapeutic decisions. As in several other studies, we found good EAs and CAs for Etest-MICs in comparison with AMB CLSI-MICs, showing values mostly between 80% and 100%, with the highest percentages observed at 24 h of reaction incubation [22,29]. Unlike Meletiadis et al., 2002 [30] and Martín-Mazuelos et al., 2003 [31], we obtained good agreement of results between the Etest and CLSI methods. Indeed, we observed increases in Etest-MIC values up to six-step dilutions for some isolates as cited in these studies [30,31].

The unique established CLSI-CBP for *A. fumigatus* is for VCZ, and the Etest performed very well, giving comparable results to the reference methodology. Otherwise, the ECV warrants improvement since the CA was unacceptably low, although Etest did reliably detect the resistance to VCZ in the single isolate of *A. fumigatus* included in this study. Very few isolates showing intermediary susceptibility profiles were correctly identified as non-WT to the agent, resulting in MEs for this commercial method. Notably, there was a high number of MiE of incorrectly identified non-WT isolates to VCZ. We should stress that an ECV corresponds to the MIC that captures ≥ 97.5% of the statistically modeled WT population and represented the probability for an isolate to be a WT isolate if its MIC was lower or equal to the ECV value. Consequently, low ECVs may overlook potentially susceptible isolates (WT), which could justify the high percentage of non-WT isolates identified by the Etest method. The poor CA observed for Etest-VCZ in comparison with the CLSI reference method suggests a need for improvement before routine employment in daily practice.

While the values for EAs remained >80% for Etest-VCZ, our MIC values for *Aspergillus* spp. tended to be lower (*p* 0.3114) by the Etest method compared with the CLSI reference method, as previously described [29,32]. The differences remained in the acceptable range of ±2 Log2-dilutions, resulting in high EA values of >90% [24,33]. For Etest and *Aspergillus* species, the best predictor of non-WT isolates, confirmed through assessments for mutants, was the proposed ECV-ITC of 2 mg/L [27]. To our knowledge, method-dependent Etest ECVs for AMB, CAS, and AND have not been extensively studied on the basis of proven mechanisms of resistance, since ECVs are based on in vitro data (either MICs or minimal effective concentration results).

There is a need to improve method-dependent ECV studies with extensive analyses in different environments and regions.

The performance of Etest in AFST with ITC and *A. fumigatus* was good in terms of EA and in CA, despite four VMEs and one ME. Previously, it was noted for *Aspergillus* spp. that the EA between ITC Etest MICs read at 24 h and reference microdilution MICs read at 48 h was 100% with RPMI agar medium, the conditions followed in this study. Conversely, for ITC, the overall agreement between Etest and M38-A for *Aspergillus* species could be as low as 67.2% [25]. This low agreement could also be due to the low reproducibility of the Etest-ITC, as previously described [30]. Although we have not studied the reproducibility, we observed, in general, MICs by the Etest to be higher, as verified in other studies [30,31,34]

Despite a substantial number of studies, including at least 25 comparative studies performed for *Aspergillus* species, with more than 3000 isolates tested against antifungal agents [22], the endpoint reading of echinocandins MECs has been found to be subjective, time-consuming, and has been associated with VME [30,35]. We confirm these issues and stress the need for studies with CAS-MICs using Etest, as well as with AND, as previously recommended [36]. The EA observed with Etest and AND was low and contrasts with published data [11,29]. In our work, the CA result for Etest and CLSI in testing CAS appeared to be not so strong in comparison with the Etest-azole results. Given the inadequate EA shown here, we could not conclude for its use in routine practice. However, in a previous study, the echinocandins EA values found for *A. fumigatus* were excellent [24].

Regarding the SYO assay for AFST of filamentous fungi, no recommendation has been released to date. Otherwise, authors have investigated the performances of the SYO to determine the MICs of filamentous fungi [10,37]. In general, excellent EA (>93.5%) with AMB, POS, or VCZ has been reported in a comprehensive review [4]. In general, the colorimetric assay performed well for *Aspergillus* species, with high overall essential agreements (≥95%) with the CLSI reference method to assess the susceptibility to triazoles and polyene drugs [10,38,39,40]. Contrary to previous data showing the lower performance of SYO for EA to test AMB [41], we found good EA, similar to Wang et al., 2018 [38]. We observed, accordingly, high categorical agreement for detecting wild type isolates, and although we verified a few MEs, SYO-AMB could not be analyzed for detection of non-WT as described previously [27], since no WT isolate was classified as non-WT by CLSI in our work.

Importantly, lower EAs for SYO-ITC than the previous (90.2–95.2%) values reported [10,31] were observed in our study, precluding us from considering SYO-ITC tests for predicting WT isolates of *A. fumigatus,* as recommended [27]. We also found unacceptable CAs due to a high level of MEs in SYO-VCZ, although SYO was able to detect the VCZ-resistant *A. fumigatus* isolate. To date, no data on such categorical agreement has been found in the literature for comparison, and the SYO method should probably not be used for routine testing in the clinical laboratory for this species/agent combination till a more feasible ECV is determined. Ideally, only one ECV should be established for each drug-bug combination. Up to now, only two multicenter studies have been performed to determine the Etest ECVs of antifungal drugs in *A. fumigatus* species and data should be rationally combined to achieve a consensus [42].

The combination of echinocandin with high-dose salvage posaconazole in cases of invasive Aspergillosis may be attempted and results from AFSTs could provide useful information for validating future CBPs or contribute to the improvement of data banks in establishing or implementing ECVs for these antifungal agents [35].

Up to now, the evaluation of the SYO for POS-MIC distributions in a large multicenter study indicated that this method provides less reliable and much lower MICs than those yielded by the CLSI method, possibly due to the different MIC determination criteria used by the laboratories [26]. We found exceptionally low CAs for SYO-POS and no literature data to compare the results. Surprisingly, an unacceptable high number of MEs were verified for POS-susceptible isolates classified as non-WT by the SYO method. The CA between results obtained by SYO with the CLSI-M38 depends on the ECV adopted. In our study, we utilized one tentative value of ECV, which may be a cause of such discrepant results. In addition, we observed a particularly good EA between results obtained by SYO and by the reference method as outlined previously [10]. Given that only very few data supported SYO MIC distributions for the tentative establishment of ECV [27], at this time, our data do not allow us to recommend its application in laboratory routines as a POS susceptibility test for *Aspergillus.* Given the importance of POS in the therapeutic arsenal for the management of invasive aspergillosis, prospective studies with larger samples of WT and non-WT isolates are necessary to establish reliable conclusions for their routine application.

Because there are no CLSI ECVs for AND and *A. fumigatus*, comparable data were not available for this study. Using the SYO method, only seven AND-MECs were readable, making impossible any robust evaluation of essential agreement. Moreover, the lack of established CLSI-ECVs and Etest-ECVs, for AND and *A. fumigatus,* prohibited the categorical agreement calculi [42,43,44]. Finally, we verified for a few isolates of *A. fumigatus* unusual phenotypes presenting trailing effects or paradoxical growth, that could create difficulties and errors in the reading of MEC endpoints. Notably, the majority of CAS as well as AND plates in the SYO experiments did not produce conclusive results given the absolute frequency of the MIC above or at the highest tested concentration (data not shown). Accepting such results led to an unreal frequency of non-WT isolates. The superior performance of the SYO plates with the other antifungal agents and correct MIC values for the control strains lead to the hypothesis of low stability of echinocandins or other factors resulting in the inactivation of these drugs. Further investigation into the usefulness of this assay for candins and *A. fumigatus* is warranted [37,43,44].

One limitation of our study is the lack of reference CBPs to determine more feasible CA percentages since the ECVs do not categorize an isolate as susceptible or resistant to certain target agents as CBPs do. Moreover, ECV carries some problems with some proposals of ECVs that still present overlapping of non-WT and WT isolates [15,16]. Another question is the small number of isolates, especially for the SYO-echinocandins tests, due to the difficulty described above in reading most reactions. Moreover, we found no previous studies regarding SYO using ITC against *A. fumigatus*, which precludes appropriate discussion of our results.

## 6. Conclusions

We have studied Etest and SYO for testing the susceptibilities of an ample collection of *A. fumigatus* clinical isolates, among other species, against first-line drugs used in the management of Aspergillosis cases. Additionally, we observed general agreement between the commercial methods for *A. fumigatus* and the CLSI reference broth microdilution reference M38-A2 method. A correlation between SYO-MICs and CLSI-MICs was demonstrated only in tests with ITC based on absolute values of MICs. More extensive studies to better assess the usefulness of SYO tests using ITC are warranted. We stress the usefulness of the colorimetric assay for detecting AMB-WT isolates. The Etest yielded EAs and CAs for testing AMB and ITC, as well as SYO for AMB. Both commercial methods presented major errors in identifying wild-type isolates as non-wild-type, capable of harboring underlying mechanisms of antifungal resistance. High EAs, but unacceptable low CAs with the SYO panel results using VCZ or PCZ were observed, thus confirming the need for better ECVs for the commercial method. Being more practical and less time-consuming for routine use, the Etest and SYO methods have potential value for the performance of susceptibility tests of *A. fumigatus*. Considering the relevance of new drugs in the therapeutic arsenal for the management of invasive aspergillosis, we recommend robust multicentric research with many WT and non-WT isolates aimed at improving these methods for further application in daily laboratory routines.

## Figures and Tables

**Table 1 pharmaceutics-14-02161-t001:** Antifungal epidemiological cutoff values and clinical breakpoints for Aspergillus fumigatus using broth microdilution, Etest and Sensititre YeastOne methods.

Method	Drug	ECV MIC (mg/L) WT<	Clinical Breakpoints MIC (mg/L)		
			S<	I	R>
Broth Microdilution	Voriconazole ^a^	ND	0.5	1	2
	Amphotericin B ^b^	2	ND	ND	ND
	Itraconazole ^b^	1	ND	ND	ND
	Posaconazole ^c^	0.5	ND	ND	ND
	Caspofungin ^b^	0.5	ND	ND	ND
Etest	Voriconazole ^d^	0.12	ND	ND	ND
	Amphotericin B ^d^	2	ND	ND	ND
	Itraconazole ^d^	2	ND	ND	ND
	Caspofungin ^e^	0.25	ND	ND	ND
Sensititre YeastOne	Voriconazole ^f^	0.25	ND	ND	ND
	Posaconazole ^f^	0.06	ND	ND	ND
	Amphotericin B ^f^	2	ND	ND	ND

ECV, Epidemiologic cutoff value; MIC, minimal inhibitory concentration; WT, Wild-Type; ND, not determined; S, susceptible; I, intermediary; R, resistant; ^a^ CLSI M61-ED2, 2020 [20]; ^b^ CLSI M59-ED3, 2020 [14]; ^c^ Espinel-Ingroff et al., 2018 [26]; ^d^ Espinel-Ingroff et al., 2019 [27]; ^e^ Espinel-Ingroff et al., 2021 [28]; ^f^ Espinel-Ingroff et al., 2019 [27].

**Table 2 pharmaceutics-14-02161-t002:** MIC (mg/L) distribution by CLSI, Etest, and colorimetric SYO methods for 106 molecularly identified clinical isolates of *Aspergillus* spp.

Specie	Antifungal Drug	Broth Microdilution Method			Etest			Sensititre Yeast One (*)		
		Range	MIC50	MIC90	Range	MIC50	MIC90	Range	MIC50	MIC90
*A. fumigatus* (*n* = 84)	Amphotericin B	0.25–2	0.5	2	<0.002–8	0.38	1.5	0.5−>8	2	2
	Voriconazole	0.12–2	0.5	1	0.094–1.5	0.19	0.5	0.12–4	0.5	1
	Itraconazole	0.06–8	0.5	1	0.06–6	1	1.5	0.06−>8	0.25	0.5
	Posaconazole	0.06–1	0.12	0.25	ND	ND	ND	0.03–1	0.12	0.5
	Caspofungin **	0.12–1	0.5	0.5	<0.002–0.19	0.064	0.125	0.008–1	0.12	0.25
	Anidulafungin **	0.001–0.03	0.003	0.3	<0.002–0.004	0.002	0.003	0.015–0.03	0.015	0.015
*A. niger* (*n* = 9)	Amphotericin B ***	0.25–1	0.5	1	0.002–1.5	0.38	1.5	2	ND	ND
	Voriconazole	0.25–1	0.5	1	1.125–1.25	1.25	0.25	0.5–1	0.5	1.0
	Itraconazole	0.06–1	1	1	0.38–2	1.5	2	0.12–0.5	0.51	0.5
	Posaconazole	0.12–0.5	0.12	0.5	ND	ND	ND	0.12–0.25	0.12	0.25
	Caspofungin	0.25–1	0.25	1	0.023–0.125	0.047	0.125	ND	ND	ND
	Anidulafungin	0.003–0.03	0.016	0.03	0.002	0.002	0.002	ND	ND	ND
*A. flavus* (*n* = 7)	Amphotericin B	0.25–2	0.12	0.25	0.19–32	0.75	32	1.0–2.0	2	2
	Voriconazole	0.25–1	0.5	1	0.125–0.75	0.19	0.75	0.25–0.5	0.5	0.5
	Itraconazole	0.06–0.5	0.25	0.5	0.5–1.5	0.75	1.5	0.12–0.5	0.25	0.5
	Posaconazole	0.06–0.25	0.12	0.25	ND	ND	ND	0.12	0.12	0.12
	Caspofungin **	0.125–0.5	0.5	0.5	ND	ND	ND	0.5	ND	ND
	Anidulafungin **	0.003–0.015	0.015	0.015	0.002	0.011	0.02	0.06	ND	ND
*A. clavatus* (*n* = 3)	Amphotericin B	0.5–1	ND	ND	0.002	ND	ND	1.0–2.0	ND	ND
	Voriconazole	2	ND	ND	1–1.5	ND	ND	0.75–1	ND	ND
	Itraconazole	0.25–1	ND	ND	0.5–1	ND	ND	0.25–0.5	ND	ND
	Posaconazole	0.12	ND	ND	ND	ND	ND	0.12–0.5	ND	ND
	Caspofungin	0.25–0.5	ND	ND	0.047–0.094	ND	ND	ND	ND	ND
	Anidulafungin	0.003	0.003	0.003	0.002–0.002	0.002	0.002	ND	ND	ND
*A. terreus* (*n* = 1)	Amphotericin B	1	ND	ND	1	ND	ND	ND	ND	ND
	Voriconazole	1	ND	ND	0.19	ND	ND	ND	ND	ND
	Itraconazole	0.5	ND	ND	1	ND	ND	ND	ND	ND
	Posaconazole	0.12	ND	ND	ND	ND	ND	ND	ND	ND
	Caspofungin	0.5	ND	ND	0.094	ND	ND	ND	ND	ND
	Anidulafungin	0.015	ND	ND	0.003	ND	ND	ND	ND	ND
*A. awamori* (*n* = 1)	Amphotericin B	0.3	ND	ND	0.19	ND	ND	ND	ND	ND
	Voriconazole	0.5	ND	ND	ND	ND	ND	ND	ND	ND
	Itraconazole	1	ND	ND	ND	ND	ND	ND	ND	ND
	Posaconazole	0.25	ND	ND	ND	ND	ND	ND	ND	ND
	Caspofungin	1	ND	ND	0.032	ND	ND	ND	ND	ND
	Anidulafungin	ND	ND	ND	ND	ND	ND	ND	ND	ND
*A. welwitschiae* (*n* = 1)	Amphotericin B	1	ND	ND	0.38	ND	ND	2	ND	ND
	Voriconazole	0.5	ND	ND	0.125	ND	ND	0.25	ND	ND
	Itraconazole	1	ND	ND	0.75	ND	ND	0.5	ND	ND
	Posaconazole	0.12	ND	ND	ND	ND	ND	0.12	ND	ND
	Caspofungin	0.5	ND	ND	0.064	ND	ND	8	ND	ND
	Anidulafungin	ND	ND	ND	ND	ND	ND	8	ND	ND

* SYO data from 58 isolates of *A. fumigatus*; ** SYO data from seven isolates of *A. fumigatus* and one isolate of *A. flavus*. *** SYO data from two isolates of *A. niger.* ND, not done.

**Table 3 pharmaceutics-14-02161-t003:** Essential agreement and categorical agreement of Etest and Sensititre YeastOne methods, compared with CLSI-M38 method, according to the established method-dependent epidemiological cutoff values (ECVs) or clinical breakpoints (CBPs), for five antifungal agents and *Aspergillus fumigatus*.

Method	Antifungal Drug	Essential Agreement (*n* Tests)	Categorical Agreement (*n* Tests; *n* of Isolates Correctly Categorized)	Type of Error	Method Dependent ECV or CBP *
Etest (*n* = 84)	AMB	82.1% (69/84)	95.2% (80/84; 80 wild types)	ME (*n* = 4)	Etest (2 mg/L) CLSI (2 mg/L)
Sensititre YeastOne (*n* = 46)	AMB	93.5% (43/46)	89.1% (41/46, 41 wild types)	ME (*n* = 5)	SYO (2 mg/L) CLSI (2 mg/L)
Etest (*n* = 55)	VCZ	87.3% (48/55)	36.4% (20/55; 10 wild type; 10 non-wild types **)	ME (*n* = 33) MiE (*n* = 2)	Etest (0.12 mg/L) CLSI (2 mg/L)
Sensititre YeastOne (n = 46)	VCZ	95.7% (44/46)	28.3% (13/46; 9 wild type; 4 non-wild types **)	ME (*n* = 32) MiE (*n* = 1)	SYO (0.25 mg/L) CLSI (2 mg/L)
Etest (n = 64)	ITC	82.8% (53/64)	90.6% (58/64; 50 wild types)	VME (*n* = 5) ME (*n* = 1)	Etest (2 mg/L) CLSI (1 mg/L)
Sensititre YeastOne (*n* = 46)	ITC	78.3% (36/46)	Not applied	Not applied	ECV for SYO unavailable
Sensititre YeastOne (*n* = 46)	POS	93.5% (43/46)	13.04% (6/46; 5 wild type; 1 non-wild types)	ME (*n* = 40)	SYO (0.06 mg/L) CLSI (0.5 mg/L)
Etest (*n* = 20)	AND	66.7% (14/21)	Not applied	Not applied	ECV for Etest unavailable

AMB, amphotericin B; VCZ, voriconazole; ITC, itraconazole; POS, posaconazole; CAS, caspofungin; AND, anidulafungin; VME, very major errors; ME, major errors; MiE, minor errors. * CLSI defined clinical breakpoints of VCZ and A. fumigatus stricto sensu: susceptible MIC ≤ 0.5 mg/L; intermediary MIC 1 mg/L and resistant MIC ≥ 2 mg/L. ** Agreement if an isolate was categorized as wild type by Etest and as susceptible by CLSI method or as non-wild type by Etest and as an intermediary/resistant by the CLSI method.

## Data Availability

The data presented in this study are available in Appendix A.

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
