# Peer review of "Evaluation of the Sensititre YeastOne and Etest in Comparison with CLSI M38-A2 for Antifungal Susceptibility Testing of Three Azoles, Amphotericin B, Caspofungin, and Anidulafungin, against Aspergillusfumigatus and Other Species, Using New Clinical Breakpoints and Epidemiological Cutoff Values"

_pharmaceutics, 2022, doi:10.3390/pharmaceutics14102161_

Round 1

Reviewer 1 Report

Melhem and colleagues investigated the reliability of antifungal susceptibility testing for Aspergillus using two commercially available assays, i.e., the Etest (solid medium based) and the YeastOne Sensititre system (microdilution based). They compared the results of > 100 included strains obtained with the two assays with a reference method (as defined by CLSI).

Some of the observed differences were surprisingly significant. However, it is quite challenging for the reader to consider those results, since the presentation of the data is not well organized and quite confusing throughout large fractions of the manuscript. While the results are worth to be published in Pharmaceutics, the organization of the manuscript and the presentation of the do not meet the quality standards of this journal.

Notably, it is surprising that a team of fourteen authors was not able to identify obvious errors (missing verbs, formatting issues [yellow marks, different font sizes, crossed out letters], typos like COVID-10, typos already in the title]) but transfers the job of lecturing the manuscript to the author.

Major comments

Introduction: The introduction needs to be seriously revised: it fills an entire page but contains as few as four references. Most statements lack any reference. Furthermore, starting from line 65 (and therefore representing the majority of the introduction) it is more a part of the material and methods section.

Materials / results: The sections 3.2, 3.3, 4.1, and 4.2 are very confusing and highly uncomfortable to read. An extensive revision of these sections would be highly beneficial for the readers.

Testing of isolates: This is quite confusing: all isolates were tested using CLSI, Etest, and SYO at LIM? A fraction of the strains was then tested (CLSI only) in Japan? Then were all strains (but only n=106 and n=60?) tested again (only CLSI) by the IAL in duplicates? And then (L149) SYO testing was performed again at LIM? And a fraction was again SYO tested by LabDIP? Later (L.209), it is stated that tests have been carried out in duplicates or triplicates. Based on which factor did the authors decide to test in duplicates or triplicates? Which results are finally presented? Have there been interlaboratory or intralaboratory differences? Why are results sometimes presented as modal and sometimes as medium value? Why do SYO results for Caspofungin and anidulafungin rely on only 7 isolates of A. fumigatus, when a total of 85 isolates (according to the materials and methods section) or 84 isolates (according to table 1…) were included? This is quite misleading, since the results are presented in a comparable manner in the same table, in which all other results were obtained by testing >80 strains.

Why did the authors “establish” different epidemiological cut off values (ECV) for antifungals for different assays, e.g., voriconazole (CLSI: 0.5, Etest 0.12, SYO 0.25)? This contradicts the basic principle of an ECV. Furthermore, are there really ECVs for all strains included in this study available, e.g., A. welwitschiae? It is stated in the results section that (using CLSI testing) “all the tested isolates were inhibited with 2 mg/L of Amphotericin B, voriconazole, and itraconazole”. But why do we get this information? The itraconazole ECV was determined by the authors (and by CLSI document M38-A) to be 1 mg/L – what is the benefit of reporting the 2 mg/L results.

Experimental design: It is stated that the conidia concentrations were 16^6 CFU/ml for CLSI and Etest testing, while the information is lacking for the SYO assay. The concentrations were determined by Neubauer chamber counting for CLSI, which is very precise, and via transmittance measurement for Etests. This is a problem for comparability: transmittance is highly dependent on the colour of the conidia, which significantly differs between different Aspergillus spp., e.g., the black conidia of A. niger vs. the yellow conidia of A. flavus. This bears the risk of different inoculum concentrations, which would have a major impact on test reliability, and which would impair comparability of the experiments. I would like to ask the authors to comment on that issue.

Minor comments

L.5: fumigatus in italic

L.6: “

L.41: tests

Ll.41-44: Please rephrase this statement.

L.44: Which were those findings of major/very major errors?

L.46: improvement of

L.53: such as s?

L.54: Which disease? Aspergillosis? COVID-10(!)? What is “(RB)” supposed to mean?

L.61: Why “although”?

L.65: Do not introduce gradient strip testing when you do not talk about it in the paragraph.

L69: Why is there a yellow marking?

Ll.78-79: “it has not been widely evaluated with commercial methods” That is not true. There have been a variety of studies, just focusing on the SYO assay, as, for instance, reviewed in 2018 by Sanguinetti and Posteraro.

Ll.88-89: There are clinical breakpoints defined for at least some drugs and some species by EUCAST, which can also be applied to commercial assays.

L.100: “Were of the highest possible grade commercially available”?

L.107: UK), The

L.121: Working with spore forming organisms, it is quite pretentious to state that all contamination can be excluded.

Ll.114-142: This section should be significantly shortened. Regarding the scope of this work, the methodology of identification is quite irrelevant.

Ll.170-171: Inoculum of up to 80 isolates of conidial suspensions?

Ll.206-207: This sentence lacks a verb.

L.229: .)when

Ll.242-245: Please rephrase this sentence.

Table 1: Why are only 84 A. fumigatus strains included in the analysis (materials and methods: 85 isolates)?

Ll.32-34: please rephrase

Style: There should be spaces between words and interjections in brackets, e.g, in the abstract in lines 37 and 38.

Author Response

September 23, 2022

Author's Reply to the Reviewer 1

Reviewer  

Melhem and colleagues investigated the reliability of antifungal susceptibility testing for Aspergillus using two commercially available assays, i.e., the Etest (solid medium based) and the YeastOne Sensititre system (microdilution based). They compared the results of > 100 included strains obtained with the two assays with a reference method (as defined by CLSI).

Some of the observed differences were surprisingly significant. However, it is quite challenging for the reader to consider those results, since the presentation of the data is not well organized and quite confusing throughout large fractions of the manuscript. While the results are worth being published in Pharmaceutics, the organization of the manuscript and the presentation of the do not meet the quality standards of this journal.

Notably, it is surprising that a team of fourteen authors was not able to identify obvious errors (missing verbs, formatting issues [yellow marks, different font sizes, crossed out letters], typos like COVID-10, typos already in the title]) but transfers the job of lecturing the manuscript to the author.

Authors:

Thank you for your valuable comments.

The manuscript was totally revised, and the English language mistakes were corrected by an expert.

Reviewer

Major comments

Introduction: The introduction needs to be seriously revised: it fills an entire page but contains as few as four references. Most statements lack any reference. Furthermore, starting from line 65 (and therefore representing the majority of the introduction) it is more a part of the material and methods section.

Authors

The introduction was revised and new references were inserted. Paragraphs were moved to the Material and Methods section.

Line 67-70 Sentence was moved to Material and Methods Section 3.2.3

The growth medium contains resazurin, an indicator of cell viability that turns from blue to pink when oxidized by viable fungi: MICs are determined after 24h of incubation as the lowest antifungal concentrations at which the wells remain blue (= no growth) and interpreted according to the CLSI breakpoints [3].

Line 85-90 to Material and Methods Section 3.2.1.

We first determined the minimum inhibitory concentration (MIC) of the routine used drugs in cases of Aspergillosis, namely three triazole agents and the polyene Amphotericin B, as well as two echinocandins, Caspofungin and Anidulafungin against 106 clinical strains of Aspergillus spp. The MIC results were then assessed to classify the isolates into different susceptibility categories, according to existing clinical breakpoints for drug-species pairs.

Reviewer

Materials / results: Sections 3.2, 3.3, 4.1, and 4.2 are very confusing and highly uncomfortable to read. An extensive revision of these sections would be highly beneficial for the readers.

Testing of isolates: This is quite confusing: all isolates were tested using CLSI, Etest, and SYO at LIM?

A fraction of the strains was then tested (CLSI only) in Japan?

Then were all strains (but only n=106 and n=60?) tested again (only CLSI) by the IAL in duplicates?

And then (L149) SYO testing was performed again at LIM?

And a fraction was again SYO tested by LabDIP?

Author:

Regarding Section 3.2: All isolates were tested by CLSI at LIM and IAL. A part of them was initially tested in Japan (CLSI)

Regarding Etest, 60 isolates were tested at IAL, among them 40 were previously tested at LIM.

SYO was used for 44 isolates at LIM and UFMS-LaDIP. Also, at UFMS-LaDIP, 17% of isolates were retested.

Section 3.3 was modified and a Figure explaining all the cutoffs was inserted to improve clarity.

Section 4.1 was summarized for giving focus on the susceptibility of A. fumigatus tested isolates to gain understanding. 

Section 4.2 was re-written. Indeed, the sentences were hardly understandable. 

Reviewer

Later (L.209), it is stated that tests have been carried out in duplicates or triplicates.

Based on which factor did the authors decide to test in duplicates or triplicates? Which results are finally presented?

Author:

We consider that replicates, repeated measurements of the same samples or technical replicates, are a valuable internal control to monitor the performance of our experiments, besides it gives accuracy to the results. The final result was the modal MIC value.

Reviewer   

Have there been interlaboratory or intralaboratory differences?

Author:

Yes, we usually observe acceptable interlaboratory and intralaboratory variations in range of +1 Log2-dilutions.

In MIC data analysis Section, we informed “All experiments were carried out in duplicate, or triplicates, and results are indicated as the modal value when distinct values were found.”

Reviewer

Why are results sometimes presented as modal and sometimes as medium value?

Author:

We apologize for the misinformation. The mean (“medium”) value was not calculated, but the mode was the adopted measure of central tendency for MIC values.  The Section 3.4 was properly corrected.

Reviewer

Why do SYO results for Caspofungin and anidulafungin rely on only 7 isolates of A. fumigatus, when a total of 85 isolates (according to the materials and methods section) or 84 isolates (according to table 1…) were included? This is quite misleading, since the results are presented in a comparable manner in the same table, in which all other results were obtained by testing >80 strains.

Author :

We are sorry for the confusing data. We did not intend to give any misleading information. Indeed, in Section 3.2.3 we informed that “Unfortunately, approximately 2/3 of the samples analysed by SYO for echinocandins showed growth in all tested concentrations (up to 8 mg/L), while for control isolates the expected MIC of CAS, and micafungin were achieved. The MIC values ≥ 8 mg/L were not registered for this study “. In the footnote of Table 1 we informed that equinocandins MIC results for A. fumigatus came from 7 isolates.

We inserted the same information in the Section 3.2.3 to clarify this issue.

Reviewer

Why did the authors “establish” different epidemiological cut off values (ECV) for antifungals for different assays, e.g., voriconazole (CLSI: 0.5, Etest 0.12, SYO 0.25)? This contradicts the basic principle of an ECV.

Author

We respect the Reviewer opinion. Otherwise, we share opinion with other authors that ECVs are dependent on the in vitro susceptibility testing method used to generate the MIC values, as follows:

  • Espinel-Ingroff et al., Multicenter study of method-dependent epidemiological cutoff values for detection of resistance in Candida spp. and Aspergillus spp. to Amphotericin B and Echinocandins for the Etest agar diffusion method. Antimicrob. Agents Chemother. 2016, 61, e01792-16.
  • Espinel-Ingroff et al. Method-Dependent Epidemiological Cutoff Values for Detection of Triazole Resistance in Candida and Aspergillus Species for the Sensititre YeastOne Colorimetric Broth and Etest Agar Diffusion Methods. Antimicrob Agents Chemother. 2018 Dec 21;63(1);
  • Salse et al. Multicentre study to determine the Etest epidemiological cut-off values of antifungal drugs in Candida spp. and Aspergillus fumigatus species complex. Clin. Microbiol. Infect. 2019, 25, 1546–1552;
  • Dellière S, Verdurme L, Bigot J, Dannaoui E, Senghor Y, Botterel F, Fekkar A, Bougnoux ME, Hennequin C, Guitard J. Comparison of the MICs Obtained by Gradient Concentration Strip and EUCAST Methods for Four Azole Drugs and Amphotericin B against Azole-Susceptible and -Resistant Aspergillus Section Fumigati Clinical Isolates. Antimicrob Agents Chemother. 2020 Feb 21;64(3):e01597-19.
  • Espinel-Ingroff et al. Etest ECVs/ECOFFs for detection of resistance in prevalent and three non-prevalent Candida spp. to triazoles and Amphotericin B and Aspergillus spp. to Caspofungin: Further assessment of modal variability. Antimicrob. Agents Chemother. 2021, 65, e01093-21
  • Espinel-Ingroff et al. Commercial Methods for Antifungal Susceptibility Testing of Yeasts: Strengths and Limitations as Predictors of Resistance. J Fungi (Basel). 2022 Mar 17;8(3):309).

Furthermore, we should not use CLSI-ECVs for interpreting MIC by commercial methods. According to CLSI M59-ED3:2020 Epidemiological Cutoff Values for Antifungal Susceptibility Testing, 3rd Edition “For ECVs to be clinically useful, the MIC or MEC should be determined using the broth microdilution procedure for yeasts (see CLSI document M272) or the broth microdilution procedure for filamentous fungi (see CLSI document M383).”

Reviewer

Furthermore, are there really ECVs for all strains included in this study available, e.g., A. welwitschiae?

Author

No, the Reviewer is right, we don’t apply the ECV for other species than A. fumigatus. In Section 3.4 we have explained “These analyses were performed for results obtained with A. fumigatus due to the robust number of isolates.”

Reviewer

It is stated in the results section that (using CLSI testing) “all the tested isolates were inhibited with 2 mg/L of Amphotericin B, voriconazole, and itraconazole”. But why do we get this information? The itraconazole ECV was determined by the authors (and by

CLSI document M38-A) to be 1 mg/L – what is the benefit of reporting the 2 mg/L results.

Author

We agree with the Review suggestion and delete the MIC information in the original manuscript.

Reviewer

Experimental design: It is stated that the conidia concentrations were 16^6 CFU/ml for CLSI and Etest testing, while the information is lacking for the SYO assay.

Author

Thank you for the correction. We insert the proper information. For the SYO assay, conidia concentration was estimated in a spectrophotometer at 530 nm adjusting to 80-82% of transmittance equivalent to 0.6-5x106UFC/mL or at Mc Farland pattern of 0.5.

Reviewer

The concentrations were determined by Neubauer chamber counting for CLSI, which is very precise, and via transmittance measurement for Etests. This is a problem for comparability: transmittance is highly dependent on the colour of the conidia, which significantly differs between different Aspergillus spp., e.g., the black conidia of A. niger vs. the yellow conidia of A. flavus. This bears the risk of different inoculum concentrations, which would have a major impact on test reliability, and which would impair comparability of the experiments. I would like to ask the authors to comment on that issue.

Author

Indeed, the pigmented conidia, especially in Aspergillus Section Nigri prevent us to use spectrometry to adjust the inoculum. In the study we evaluate MIC for isolates from distinct species by CLSI using Neubauer chamber to this procedure. For Etest and SYO using A. niger we employed the same tool (We inserted this information in the revised version of the manuscript). For the remaining species we were able to carefully adjust the density after training when we standardized the density of A. flavus suspension after colony counting. Beyond that we observed the replicate results.

Reviewer

Minor comments

L.5: fumigatus in italic

Author

Thank you for the correction, we changed it to italics.

Reviewer

L.6: “

Author. We appreciate, the mark was removed.

Reviewer

L.41: tests

Author

Thank you, the word was corrected.

Reviewer

Ll.41-44: Please rephrase this statement.

For SYO, tests with AMB, VCZ, and POS result in excellent essential agreement (>93% MICs within ± 2 log2 dilutions), and >80% for AMB, VCZ and ITC Etest-MICs High categoric agreement (>85%) was found for AMB, ITC and CAS Etest-MICs and for SYO-AMB (>90%).

Author

We have revised and improved the sentences: “For SYO, tests with AMB, VCZ, and POS result in excellent essential agreement (>93% MICs within ± 2 log2 dilutions), and >80%. for Etest-MICs for AMB, VCZ and ITC. High categoric agreement (>85%) was found for Etest -MICs for AMB, ITC and CAS Etest-MICs and (>90%) for SYO-AMB.”

Reviewer

L.44: Which were those findings of major/very major errors?

Author

According the Table 2, we found major/very major errors in MIC results of azole drugs and Amphotericin B, in distinct proportions for SYP and Etest method. We were not able to explain more about these errors in Abstract Section due to the maximum characters number. We tried to improve the information otherwise: “ The considerable number of major/very major errors found using Etest and SYO, possibly related to the proposed cutoffs, associated with the practical and less time-consuming characteristics and potential value, support the need for improvement of commercial methods for use with Aspergillus strains.”.

Reviewer

L.46: improvement of

Author

Thank you, the preposition was added.

Reviewer

L.53: such as s?

Author

Sorry, we rewrote the sentence.

such as the increasing number of patients who undergo organ transplants [1]

Reviewer

L.54: Which disease? Aspergillosis? COVID-10(!)? What is “(RB)” supposed to mean?

Author

We replaced the word disease with Aspergillosis; COVID 10 by COVID 19 and (RB) with a reference.

Reviewer

L.61: Why “although”?

Author

We change the sentence:

AFST plays a significant role in patient management by aiding the prescription of an appropriate antifungal, however the reference methodologies…

Reviewer

L.65: Do not introduce gradient strip testing when you do not talk about it in the paragraph.

Author

Thank you for the suggestion about this section. This line and other were moved to Material and Methods Section as already described.

Reviewer

L69: Why is there a yellow marking?

Author

Please do not consider. We excluded it. And moved this sentence to Methods section.

Reviewer

Ll.78-79: “it has not been widely evaluated with commercial methods” That is not true. There have been a variety of studies, just focusing on the SYO assay, as, for instance, reviewed in 2018 by Sanguinetti and Posteraro.

Author

We appreciate the suggestion and indeed, we found two additional articles (Mello et al, 2017; Siopi et al., 2017) following your recommendation to improve Discussion Section.

Mello E, Posteraro B, Vella A, et al. Susceptibility Testing of Common and Uncommon Aspergillus Species against Posaconazole and Other Mold-Active Antifungal Azoles Using the Sensititre Method. Antimicrob Agents Chemother. 2017;61(6):e00168-17. Published 2017 May 24. doi:10.1128/AAC.00168-17

Siopi M, Pournaras S, Meletiadis J. Comparative Evaluation of Sensititre YeastOne and CLSI M38-A2 Reference Method for Antifungal Susceptibility Testing of Aspergillus spp. against Echinocandins. J Clin Microbiol. 2017;55(6):1714-1719. doi:10.1128/JCM.00044-17

Reviewer

Ll.88-89: There are clinical breakpoints defined for at least some drugs and some species by EUCAST, which can also be applied to commercial assays.

Author

Indeed, but as we performed CLSI broth dilution method we think it is better to adopt the single clinical breakpoint defined by the Institute CLSI.

Reviewer

L.100: “Were of the highest possible grade commercially available”?

Author

We used:
Itraconazole or Voriconazole ≥98% (TLC or HPLC)

Amphotericin B 80% (HPLC)

Caspofungin or Anidulafungin  ≥97% (HPLC)

Reviewer

L.107: UK), The

Author

(The) was removed.

Reviewer

L.121: Working with spore forming organisms, it is quite pretentious to state that all contamination can be excluded.

Author

We deleted the sentence, although we work at certificated laboratories.

Reviewer

Ll.114-142: This section should be significantly shortened. Regarding the scope of this work, the methodology of identification is quite irrelevant.

Author

We tried to short the information on identification procedures.

Reviewer

Ll.170-171: Inoculum of up to 80 isolates of conidial suspensions?

Author

We procedured up to 106 tests for Amphotericin B by Etest method. We corrected the sentence as “Inoculum of up to 106 isolates of conidial suspensions were successively...”

Reviewer

Ll.206-207: This sentence lacks a verb.

Author

We rephrased as: Regarding A. fumigatus and AND, neither CBP nor ECV was defined by CLSI or elsewhere. For the remaining drugs we adopted the existing CBP or ECV values as follows:

Reviewer

L.229: .)when

Author

.) was deleted.

Reviewer

Ll.242-245: Please rephrase this sentence.

Author

Thank you for the critics, indeed the sentence needed a correction. We proposed: “The CLSI-MICs for QC strains were in the established reference MIC/MEC ranges recommended [11].”

Reviewer

Table 1: Why are only 84 A. fumigatus strains included in the analysis (materials and methods: 85 isolates)?

Author

Sorry for the misinformation, the correct number is 84 isolates of A. fumigatus.

Reviewer

Ll.32-34: please rephrase

Author

We improved the idea: “In suspected or proven azole-resistant A. fumigatus cases, AMB remains the first-line therapy and a reliable simple commercial AST is needed to timely provide the fungal susceptibility profile to help with therapeutic decisions.”

Reviewer

Style: There should be spaces between words and interjections in brackets, e.g., in the abstract in lines 37 and 38.

Author

Thank you. These changes were made in the text.

Reviewer 2 Report

The antifungal susceptibility testing of  clinically relevant Aspergillus isolates  represent a challenge for routine diagnostics laboratories. Despite of availability of reference protocols as well as a few commercial tests evaluated in several studies there is no consensus about their utility in practices. Presented study try to respond to this need. The study is well written and the data properly analysed. The number of isolates was 106, what is quite high. The disadvantage is that susceptibility to particular antimicotics by Etest and SYO was tested on  lower number of isolates. And the second, is that the number of resistant isolates was low. Moreover, the resistance was not determined/confirmed molecularlly.

Author Response

Dear Reviewer,

I appreciate your comments and suggestions.

Unfortunately, the rarity of resistance to Amphotericin B prevents us from evaluating them. In our big culture collection containing more than 600 Aspergillus strains, we have no such strain to include in the study. Otherwise, we included one VCZ resistant isolate plus 4 non-wild-type isolates for POS (n = 1), for CAS (n= 1) and for ITC (n = 2).   As explained in Section 3.2.3 we acquired a considerable number of SYO tests, but most plates were discharged in view of our scale results. The cost of Etest strips prevents us to acquire material for all strains. So, we prioritized Amphotericin B tests due to the therapeutic choice in routine. Moreover, we decided to replicate the tests to improve the quality and accuracy of the results.

 Regarding the search for underlying mechanisms of resistance, we are conducting another study performing CYP51A sequencing. Still, we haven’t finished the analyses due to the laboratory difficulties faced during the COVID-19 pandemic, related to human resources and access to the Sanger facilities. Meanwhile, all people and equipment are restricted to SARS-Cov 2 examens, besides Monkey Pox outbreaks.

Reviewer 3 Report

Invasive aspergillosis is an infectious diseases associated with high mortality.

In addition to rapid and accurate diagnosis, reliable susceptibility testing is necessary for adequate treatment to the patients allowing successful recovery.

Standardized tests to determine the antifungal susceptibility of Aspergillus isolates are time-consuming and laborious, therefore, it is extremely important to evaluate alternative commercial tests that are reliable and reproducible for use in routine laboratories.

This work evaluated the performance of 2 commercial tests, Etest and SYO with 106 clinical Aspergillus isolates with respect to 3 azoles, AMB and 2 echicandins in comparison with the reference test CLSI-M38-A2.

Among the Aspergillus species studied, 84 A. fumigatus isolates were tested, which supported the results obtained.

The methodology is well described and the results are presented clearly and adequately. The results are well discussed and compared with the current available literature.

This work is important because it contributes to the interpretation of the Etest and SYO methodologies to Aspergillus isolates susceptibility to the antifungals tested. However, as well stated by the authors, more Aspergillus isolates  should be tested in multicentric collaborations for improve security in the interpretation of these 2 commercial tests.

Minor corrections:

Corrections with native English translator is very important to give greater clarity to the text.

Author Response

Dear Reviewer,

We appreciate your consideration.

We agree that future multicentric studies will give more information about the usefulness of these two commercial tests in clinical practice.

Reviewer

Minor corrections:

Corrections with a native English translator is very important to give greater clarity to the text.

Author

Thank you. The manuscript has been revised by an expert English translator.

Round 2

Reviewer 1 Report

Thank you for your interesting work!